# The Effectiveness of Topical Dimethicone Together with a One Health Approach for the Control of Tungiasis in the Sanumás Communities, Yanomami Territory, Amazon Rainforest: A Real-World Study

**DOI:** 10.3390/tropicalmed8110489

**Published:** 2023-10-30

**Authors:** Yago Ranniere Teixeira Santana, Débora Dornelas Belchior Costa Andrade, Daniel Holanda Barroso, Andressa Vieira Silva, Lucas Felipe Carvalho Oliveira, Renata Velôzo Timbó, David Dias Araújo, Rafael Rocha de Andrade, Marcos Antonio Pellegrini, Fabiola Christian Almeida de Carvalho, Luciana Pereira Freire Martins, Ciro Martins Gomes

**Affiliations:** 1Programa de Pós-Graduação em Ciências Médicas, Universidade de Brasília, Brasília 70910-900, Brazil; yago.santana@saude.gov.br (Y.R.T.S.); debora.dbca@gmail.com (D.D.B.C.A.); danielhbarroso@gmail.com (D.H.B.); andressa.dermatologia@gmail.com (A.V.S.); renata.timbo@gmail.com (R.V.T.); rafaelandrade@unb.br (R.R.d.A.); 2Secretaria Especial de Saúde Indígena (SESAI), Ministério da Saúde do Brasil, Brasília 70723-040, Brazil; lucasfelipe250@hotmail.com; 3Programa de Pós-Graduação em Saúde Coletiva, Universidade de Brasília, Brasília 70910-900, Brazil; 4Dimensuri Assessoria Técnica, Brasília 70391-900, Brazil; daviddiasaraujo@gmail.com; 5Programa de Pós-Graduação em Ciências da Saúde, Universidade Federal de Roraima, Boa Vista 69304-000, Brazil; marcos.pellegrini@ufrr.br (M.A.P.); fabiola.carvalho@ufrr.br (F.C.A.d.C.); 6Programa de Pós-Graduação em Patologia Molecular, Universidade de Brasília, Brasília 70910-900, Brazil; freire_luciana@yahoo.com.br

**Keywords:** tungiasis, Amerindian People, One Health, neglected diseases

## Abstract

Background: The success of tungiasis treatment is highly dependent on adequate environmental control. Methods: This is a real-world observational cohort study designed to monitor the effectiveness of topical dimethicone together with a One Health approach for the control of tungiasis in the Sanumás communities, Amazon rainforest, Brazil. We followed up on 562 indigenous people and 81 domestic dogs for 1.5 years in a 3-month interval. A new molecular method for large-scale soil evaluation was also tested. The control of tungiasis was independently conducted by the Brazilian Ministry of Health and comprised topical dimethicone application (NYDA^®^) for humans, single-dose oral afoxolaner for dogs, and in-house soil fumigation with fipronil. The main outcome was the occurrence of tungiasis after the use of topical dimethicone together with the One Health approach. Results: A total of 49 of the 562 indigenous people had active tungiasis at enrollment (8.72%). Only three cases of tungiasis resulted in active lesions after the use of topical dimethicone together with the One Health approach, with two cases of recurrence. From the 6-month follow-up and after, soil infestation was not detected. Conclusions: We conclude that the use of NYDA^®^ together with animal and environmental interventions are effective measures for the control of tungiasis.

## 1. Introduction

Tungiasis is an ectoparasitosis caused by *Tunga penetrans* (Insecta, Siphonaptera, Tungidae). The pathogen is native to Central and South America but is also endemic to Africa [1,2,3]. These parasites live in the dry soils of rural areas and in communities with low socioeconomic development [4]. Control can be difficult in communities where *T. penetrans* infect humans and animals and where they can be abundantly found in the soil. As a result of this complex environmental interaction, the disease shows epidemic characteristics in some parts of sub-Saharan Africa and South America [5,6].

The Yanomami territory is a 192,000 km^2^ area located in northern Brazil and Venezuela. The Sanumás are a subgroup of the Yanomami linguistic family [7] and are divided into different communities that are settled along river courses and in places with plain terrain. The geographical characteristics of this region have been described elsewhere (Access point: (4°00′1″ N 64°29′22″ W; altitude = 759 m above sea level)) [6]. A recent survey showed that the point prevalence of tungiasis in this community was 8.11% in January 2022 (95% confidence interval = 6.04–10.78%) [6]. Additionally, the disease mainly affects children in the Sanumás communities and can result in severe secondary complications, including verrucous coccygeal lesions, secondary infection, and deformities (Figure 1) [6]. Moreover, *T. penetrans* is widely present in the in-house soil of residences, and this is the most important risk factor for the occurrence of tungiasis [6]. This population lives in conditions of low social development. Houses are made of straw or clay, sometimes without walls, and are built on compacted natural soil [6].

The occurrence of tungiasis in a remote area such as the Sanumás territory resulted in a public health problem detected in the late 2010s. In 2018 and 2019, the World Health Organization (WHO) and the Brazilian Ministry of Health initiated joint efforts to control tungiasis in the indigenous communities of South America, mainly in Brazil and Colombia. Those efforts comprised proposing a cost-effective One Health approach that aims to simultaneously treat humans, animals, and the environment, seeking optimal conditions through a collaborative and continuous set of actions [8,9]. In 2019, the Brazilian government recruited and financed specialized dermatology professionals to solve this important public health problem.

This study aimed to evaluate the effectiveness of topical dimethicone together with a One Health approach, comprising animal and environmental measures for the control of tungiasis using a real-world cohort follow-up strategy in a recently contacted indigenous community located in the Amazon rainforest. Special attention was given to factors that might explain reinfection in treated indigenous people.

## 2. Materials and Methods

This real-world study was designed to monitor the effectiveness of topical dimethicone together with a One Health approach, comprising animal and environmental actions for the control of tungiasis in the Brazilian Amazon rainforest. As tungiasis is a neglected disease, there are no drugs or supplies registered for specific treatment in most endemic countries. However, for indigenous communities, the Brazilian Ministry of Health accesses specific healthcare supplies donated by the WHO. The Brazilian indigenous populations frequently affected by tungiasis are protected by special national laws, and conducting a controlled clinical trial is often unfeasible. Brazil has one of the largest populations of native ethnic groups that live in protected forest areas [10].

Considering this scenario, from 2019 to 2023, the Brazilian Ministry of Health financed all the materials considered necessary for the control of tungiasis, as needed by the Distrito Sanitário Especial Indígena (DSEI) Yanomami, which is responsible for indigenous health in the Yanomami territory. Our research team was responsible for real-world observational monitoring, aiming to assess the effectiveness of topical dimethicone together with this One Health approach. The researchers had no influence on the decision to apply human, animal, or environmental measures. We examined all the inhabitants living in the six Sanumás settlements (Karanaú, Kulapoipú, Psicultura, Katarrinha, Caixa D’Água, and Kiripassipu). Previous research protocols that described the point prevalence of tungiasis and other diseases in this population were indispensable for carrying out a long-term therapeutic follow-up and are published elsewhere [6,11].

The therapeutic actions began on 26 January 2022, together with our observational research procedures. Human, animal, and environmental treatments were reapplied every 3 months if necessary, according to the presence of human/animal tungiasis or according to the presence of *T. penetrans* in in-house soils. From the third monitoring visit, soil treatment was no longer necessary due to the absence of active *T. penetrans* (see Section 3).

The therapeutic actions comprised three measures performed on the same day and were repeated every 3 months at the discretion of the assistant team.

### 2.1. Human Measures

All indigenous people diagnosed with tungiasis received a 1-cycle topical application of dimethicone in the form of NYDA^®^ (Pohl-Boskamp GmbH& Co. KG, Hohenlockstedt, Germany) [12,13]. According to the manufacturer, NYDA^®^ is composed of two main compounds, one volatile and one viscous. Volatile dimethicone facilitates product application, and the viscous compound occludes the respiratory openings of the parasites. After a gentle cleansing of the lesions with water and soap, a certified nurse applied three to six drops of the medication to each affected area, and the medication was spread with the aid of a gloved finger. Each patient received three equal applications with an interval of 10 min between each application. The therapeutic team used occlusive dressings for 24 h after topical application to the feet of indigenous people who complied with and agreed to this procedure. However, most indigenous people did not agree with the use of occlusive dressing, claiming that it might interfere with daily activities.

### 2.2. Animal Measures

During home visits, indigenous people were questioned about the presence of domestic animals, and all domestic dogs with weights ranging from 4 to 10 kg received a single dose of 1.25 g of afoxolaner (NexGard^®^, Boehringer Ingelheim, Paulínia, São Paulo, Brazil). No other domestic animals were found. Despite previous reports that Sanumás raised animals such as pigs, cows, and chickens, these animals were not present in the communities assessed. A serial evaluation of all domestic animals proved to be unfeasible. The animals’ living conditions require them to travel long distances daily in search of food.

### 2.3. Environmental Measures

After house cleansing with proper brooms and the implementation of human protection measures, the household floor and the lower part of the wall of the houses, together with the ground half a meter away from the wall, was fumigated using a small quantity of 800 g/kg carbonitrile (FIPRONIL) 80% m/m (UPL do Brasil Indústria e Comércio de Insumos Agropecuários S.A., Ituverava, Brazil), diluted (1 g/L) in clean water. The amount applied was not to soak the soil but rather cover the entire target area. Spraying was only performed if live fleas were present, according to a soil evaluation method described elsewhere [6]. For safety reasons, the health conditions of the population and soil and water monitoring are regularly realized by the Brazilian Ministry of Health.

### 2.4. Recruitment and Follow-Up Protocol

The 1-year recruitment period started on 26 January 2022, following a previous study that assessed the point prevalence of tungiasis in the target population, and was resumed on 5 February 2023 [6]. Recruitment visits were at 3-month intervals. The first three visits were made by a specialized team from the University of Brasília, and the remaining visits were made by trained local team members who directly reported to the university team and to the Brazilian Ministry of Health.

The 79-week (1.5 years) follow-up visits were every 3 months, starting on 26 January 2022 and ending on 30 July 2023 (Figure 2). Similar to the recruitment protocol, the first three visits were made by a specialized team from the University of Brasília, and the remaining visits were made by trained local team members. The observational procedures comprised human, animal, and environmental evaluations, which are described in detail in the next section.

The first three observation visits were made by a team comprising a certified dermatologist, a primary care physician specializing in indigenous health, an entomologist, a nurse, a nursing technician, and a local indigenous health agent (interpreter). Subsequent visits were made by a trained local team comprising a primary care physician, a nurse, and a nursing technician, as monitored by professionals from the Brazilian Ministry of Health. A case of tungiasis was diagnosed according to clinical criteria. Tungiasis severity was classified as follows: 1—mild (fewer than 10 active lesions limited to the plantar region); 2—moderate (fewer than 10 active lesions in the plantar and palmar regions); and 3—severe (more than 10 lesions, usually affecting the coccygeal region) [6].

Animal evaluation was performed by a certified veterinarian in the first three visits; the remaining evaluations were performed by a trained nursing technician. Entomologists were also responsible for direct soil evaluation and for soil sample collection for molecular analysis in the first three visits.

### 2.5. Molecular and Direct Evaluations of Soil Infestations

We used two methods for the evaluation of soil infestations from internal areas of the houses: 1: direct and microscopic examination methods and 2: molecular examinations based on a quantitative real-time-based polymerase chain reaction (qPCR). All the procedures are explained in detail below.

### 2.6. Direct and Microscopic Examination Methods

As previously described [6], depending on the size of each residence, a certified entomologist placed three to five sheets of white paper on in-house soil. A light scraping of the soil with a sharp instrument was carried out around each sheet. After a few seconds of rubbing against the ground, fleas began to jump on the sheets of paper and could be counted. After 3 min, the entomologist performed a visual count using a Dermlite 2 Hybrid dermoscope (Dermlite, San Juan Capistrano, CA, USA) (Figure 1). The presence of adult *T. penetrans* in soil was quantified as follows: 0 = no infestation (no sheet with parasites), 1 = light infestation (only one sheet with parasites), 2 = medium infestation (two sheets with parasites), and 3 = severe infestation (three or more sheets with parasites) [6]. A microscopic analysis of each 30 g soil sample was performed at the Dermatology Laboratory, UnB, Brazil, for species classification.

### 2.7. Molecular Examinations

Genetic sequences of *T. penetrans* were obtained from the GenBank database (https://www.ncbi.nlm.nih.gov/genbank/ (accessed on 1 August 2023)) using accession numbers EU169194, EU169196, and EU169197; these sequences represent repetitive sequences in the internal transcribed spacer (ITS1) region of the ribosomal DNA of *T. penetrans* [14]. Primer pairs were designed for the amplification of *T. penetrans* DNA using the Primer BLAST tool (https://www.ncbi.nlm.nih.gov/guide/howto/design-pcr-primers/ (accessed on 1 August 2023)). The reaction was validated using the primer pair 5′-TAATCCCGGTAACGGGTGCT-3′ and 5′-CCACCAGTGATCCACCGTTC-3′, specific for *T. penetrans* EU169196.1 and EU169193.1. To establish a positive control, a *T. penetrans* target gene was cloned using an *Escherichia coli* strain with the Geneart tool (Thermo Fisher, Waltham, MA, USA). The synthetic gene (gene size 945 base pairs) was assembled from synthetic oligonucleotides and/or PCR products. The fragment was inserted into pMK-RQ (KanR). The plasmid DNA was purified from the transformed bacteria, and the concentration was determined via UV spectroscopy. The final construct was verified by sequencing. The sequence identity within the insertion site was 100%.

For quantification, a standard curve was established using the following protocol (Figure 3) (R^2^ = 0.959; slope = −3.251; efficiency = 103.065). The standard curve used cloned DNA at five dilution points starting at 1 million gene copies up to 100 gene copies. Reactions were performed with a QuantStudio 1 instrument (Applied Biosystems, Foster City, CA, USA) and PowerUp SYBR Green Master Mix (Applied Biosystems, Foster City, CA, USA), with a final volume of 15 µL. Each reaction contained 1x Universal SYBR Green PCR master mix (Applied Biosystems, Foster City, CA, USA), both 10 µM primers (Applied Biosystems, Foster City, CA, USA), and 3 µL of ultrapure water to the final volume. Amplifications were performed as follows: an initial step at 50 °C for 2 min, 95 °C for 2 min, and 45 cycles at 95 °C for 15 s and 60 °C for 1 min. A melting curve was constructed using 0.3 °C increments from 55 °C to 95 °C (Figure 3). Soil samples were collected and stored in 1.5 ml microtubes at room temperature and processed in the Dermatology Laboratory, University of Brasília, Brazil. DNA extraction occurred at a maximum of 15 days after collection. DNA extraction from the soil collected near in-house fireplaces was performed using 200 mg of soil near the house fireplace with a PureLink™ Microbiome DNA Purification Kit (Thermo Fisher, Waltham, MA, USA).

### 2.8. Risk Factors for Tungiasis Persistence

According to a previously published point prevalence study [6], the main risk factors for tungiasis occurrence in the Sanumás territory are age group, number of houses in the community, number of households in the community, number of inhabitants per house, soil infestation severity, number of dogs belonging to the household, and a previous personal or family history of tungiasis. These factors were considered the main risk factors for tungiasis occurrence in our real-world cohort strategy.

### 2.9. Outcome Evaluation

Before starting any cohort study, it is important to define the most important outcome. This will be used for the most important study procedures, such as clinical assessment and sample size calculation. We chose a dichotomous outcome for its simplicity of analysis. The main outcome for our real-world cohort evaluation was the occurrence of tungiasis (yes or no) after the use of topical dimethicone together with the One Health approach initiated at the first visit of our observational study.

This main outcome was evaluated at two main time points, classified as short-term follow-up and long-term follow-up. Based on previous clinical trials, we postulated that dimethicone (NYDA^®^) [15] application would have a greater effect on early outcomes, mainly interfering with the cure of existing tungiasis lesions, but that animal and soil treatment would probably have a greater effect on later outcomes and reinfection—an outcome rarely evaluated in previous clinical trials [15]. The first time point (short-term follow-up) was set at 7 days after the first application of topical dimethicone together with the One Health approach for selected indigenous people living in the communities of Karanaú, Kulapoipú, Psicultura, and Kiripassipu [6] and aimed to evaluate the short-term action of topical dimethicone. The second timepoint was set at 79 weeks (1.5 years) (long-term follow-up) after the first application of topical dimethicone together with the One Health approach and was designed to evaluate tungiasis-free survival and tungiasis recurrence at every 3-month visit.

### 2.10. Sample Size

Sample size calculation was based on two main objectives: 1: to assess the association of possible risk factors for tungiasis recurrence and 2: to assess the period prevalence of tungiasis. Sample size calculation was performed before patient recruitment in accordance with our research protocol registered at the Brazilian national site Plataforma Brasil (https://plataformabrasil.saude.gov.br/ (accessed on 1 August 2023)).

The first and main objective was to monitor the occurrence of tungiasis after treatment. For this purpose, the likely effect of predictors such as the type of treatment needed to be estimated before recruitment began. To identify possible associations, a bilateral significance of 99%, a power of 80%, and a proportion of 1 case to 1 control were established. We considered a reduction of 10% in the prevalence of tungiasis sufficient to assess most characteristics after the definition of cutoff points to generate dichotomic variables, including age, house soil infestation level, number of houses in the community, number of inhabitants in the same house, and number of domestic animals living in the house. Finally, we attributed a 15% outcome frequency for indigenous people exposed to the referred risk factors and a 5% outcome frequency for indigenous people not exposed to the referred variable. A minimum sample size of 160 indigenous people with and 160 indigenous people without the variable of interest was necessary to test the effects of the variable on the occurrence of tungiasis.

The sample size calculation for prevalence studies is only relevant when the entire target population (Sanumás) cannot be assessed and when a randomized sample will be assessed. This was the initial intention of the study, which proved to be unfeasible. Regarding prevalence, we overestimated the prevalence of tungiasis in the population at 50% of the expected Sanumás population (3000 individuals), because previous data are scarce and can vary according to migration waves. Considering a confidence limit of 5% and a design effect of 1, the minimum sample size was calculated at 544 indigenous people to achieve a 99% confidence interval. Recruitment randomization was not feasible due to the large distances between the houses in the communities; therefore, we opted to evaluate all indigenous people in the communities previously found to have the highest infection rates according to Secretaria Especial de Saúde Indígena (SESAI), Brazilian Ministry of Health (Karanaú, Kulapoipú, Psicultura, Katarrinha, Caixa D’Água and Kiripassipu) [6]. This strategy allowed for precise prevalence measurements in those specific communities but might result in an overestimation of cases for the total Yanomami region. The sample size calculation was performed using OpenEpi, version 3.01 (Emory University, Rollins School of Public Health, Atlanta, GA, USA).

### 2.11. Statistical Analysis

Univariate comparisons of categorical variables were performed using the chi-square test or Fisher’s exact test, depending on the frequency of each occurrence. Regarding age, indigenous people were divided into only two groups: children and adults, because this information was not reliable. We performed the *t* test or the Wilcoxon test for numerical variables depending on the nature and distribution of the data.

The results of therapy with topical dimethicone were evaluated at 7 days (short-term follow-up) and 1.5 years (long-term follow-up) after the application procedure. For the short-term and long-term follow-up timepoints, we performed only descriptive analysis of the outcome (tungiasis occurrence after the One Health approach). The relative risk calculation and multivariate analysis were unfeasible due to the low number of indigenous people who presented tungiasis recurrence. A comparison of the agreement of positivity or negativity of the direct and molecular evaluation of the soil of each house was performed using McNemar’s test. The infestation of indoor soil was defined by the finding of any number of adult fleas in the direct examination or by the positivity of the molecular evaluation with qPCR.

Statistical analysis was performed using the program R version 4.1.2 (R Core Team (2021)). R: A language and environment for statistical computing. R Foundation for Statistical Computing, Vienna, Austria. URL https://www.R-project.org/ (accessed on 1 August 2023)). Statistical significance was defined as a *p* value < 0.05 and a 95% confidence interval (CI).

### 2.12. Ethical Aspects

Indigenous people were enrolled after written informed consent was obtained from all subjects involved in this study or their legal guardians. This research complies with all international and national ethics exigences as well as with Brazilian laws designed to protect vulnerable indigenous communities. This study was approved by the Ethics Committee of the Faculty of Medicine (UnB) (Reference code CAAE: 30638920.0.0000.0008), by the Comissão Nacional de Ética em Pesqusisa (CONEP), by the Conselho Distrital de Saúde Indígena (CONDISI) (a sector of the DSEI formed by native indigenous leaders), and by the leadership of each community. Access by the research team to the restricted area of the DSEI Yanomami was made possible via authorization of SESAI. The team accessed the community using a Cessna 208 “Caravan” (Cessna, Wichita, KS, USA) from the city of Boa Vista, Brazil.

## 3. Results

### 3.1. Demographic, Environmental, and Sanitation Evaluations

We evaluated 562 indigenous people living in 79 houses. A total of 45 cases of tungiasis were identified in 555 indigenous people living in 78 houses at the first visit in January 2022. Detailed epidemiological characteristics of this first visit are described elsewhere in a point prevalence study [6]. In the third trimester evaluation, in July 2022, a family from the same ethnic/linguistic group had migrated to a new house in the Karanaú community from Venezuela. There were seven new residents (two adults and five children) from this family; four of the indigenous people had arrived with a diagnosis of tungiasis, representing nonautochthonous cases. In addition to the 79 dogs (68 with tungiasis) evaluated in January 2022 [6], this family had two dogs that were not infected by *T. penetrans.*

### 3.2. Factors Associated with Tungiasis Occurrence at the Time of Patient Inclusion

Following the real-world cohort protocol, all indigenous people were included for basal characteristics analysis. Details of this population comprising 562 indigenous people are provided in Table 1.

### 3.3. Tungiasis Case Classification

Of the 49 cases of tungiasis detected at the recruitment phase, 18 were classified as mild, 20 as moderate, and 11 as severe. All indigenous people manifested plantar lesions; 20 also had associated hand lesions, and 11 also had coccygeal lesions. All indigenous people who presented with coccygeal lesions also developed active palmar and plantar lesions.

### 3.4. Short-Term Follow-Up

We evaluated the effects of the described therapy at 7 days after the first baseline evaluation in January 2022 to provide a direct evaluation of the effect of topical dimethicone in humans. Analyses were conducted for the communities of Karanaú, Kulapoipú, Psicultura, and Kiripassipu, comprising 362 indigenous people in total (convenience sample); among them, 26 cases of tungiasis were treated. Four indigenous people had mild tungiasis, 14 had moderate tungiasis, and eight had severe tungiasis. Regarding patient treatment areas, we evaluated lesions in 57 areas, including 27 plantar lesions, 22 palmar lesions, and eight coccygeal lesions.

In six indigenous people with six plantar lesions, occlusive dressing was applied for 24 h after dimethicone application, and none of the indigenous people presented active lesions 7 days later (Figure 4 and Figure 5). A unique active lesion remained on the left foot of only one patient with moderate tungiasis in the plantar and palmar areas who did not use occlusive dressings. All remaining lesions were in the involutionary phase. No side effects were reported.

A reevaluation of the treated animals 7 days after afoxolaner administration and soil evaluation 7 days after fipronil application were not possible.

### 3.5. Long-Term Follow-Up

In the long-term follow-up, we observed that the application of topical dimethicone together with the applied One Health approach was highly effective (Figure 6 and Figure 7). Only three cases of tungiasis were detected after the first study visit when the One Health approach was first applied in January 2022. Two indigenous people who were treated for tungiasis at the first visit presented mild tungiasis recurrence at 13 and 52 weeks after the first visit. At the basal evaluation, these indigenous people presented moderate or severe tungiasis. Additionally, one patient who did not present with tungiasis at the first study visit was diagnosed with mild tungiasis at 26 weeks of follow-up. The remaining indigenous people remained free of tungiasis during the 1.5-year follow-up.

In the first visit that occurred in January 2022, the internal soil of 60 of the 78 (75.64%) houses evaluated were infested according to the direct (median infestation level = 1) [6] and molecular evaluation (mean infestation level = 1757.46 *T. penetrans* gene copies/200 mg of soil). In the second visit that occurred in April 2022, the internal soil of only 22 of the 78 houses (28.21%) was infested according to the direct (median infestation level = 0) and molecular evaluation (mean infestation level = 37.72 *T. penetrans* gene copies/200 mg of soil). In the soil evaluation, in addition to *Tunga penetrams,* we found a massive infestation by *Ctenocephalides felis* that also infested domestic dogs, according to the veterinary evaluation. Although direct on-site examination is not capable of differentiating species, our molecular target was specific.

From the third study visit, which occurred in July 2022, and in subsequent visits, soil infestation was not detected, even in the house of the family of migrants from Venezuela, although cases of human tungiasis occurred in that specific house. The comparison between the direct and molecular evaluations of the soils of each house showed that the results of the methods did not agree in most cases in the first (*p* < 0.001) and second (*p* = 0.004) quarterly visits.

Monitoring the animals was very difficult due to the dogs’ high mobility. On the second visit, only 26 dogs seen on the first visit were found, and none of them showed any sign of tungiasis. On the third visit, only 18 dogs were found, and none had any signs of active tungiasis. In subsequent visits, the tracking of dogs seen at recruitment was not feasible, but there were no subsequent records of dogs with active tungiasis in the community. No adverse events were reported in the dogs, and the community accepted single-dose oral therapy very well.

## 4. Discussion

This study shows that the control of tungiasis in communities where the disease is endemic is a very specific issue that requires very specific measures. This disease generally occurs in underdeveloped environments where healthcare access is difficult. The Sanumás communities are located in Brazil’s extreme north and are not accessible by land or river transportation [6]. Overall, the lack of continuous health care is not only a risk factor for tungiasis but can jeopardize any traditional control measures.

In locations where tungiasis is endemic, there is an established interaction between human, animal, and environmental conditions to which *T. penetrans* has already adapted [16]. During recruitment, 49 cases of tungiasis were diagnosed among the 562 indigenous people evaluated. Three other cases were detected at follow-up visits, but only one had no previous history of tungiasis. A total of 50 indigenous people (49 during recruitment phase + one during the follow-up period) had at least one episode of active tungiasis during the follow-up period (8.89%, 95% CI = 6.68–11.56%), similar to the estimated point prevalence previously described for this population (8.11%, 95% CI = 6.04–10.78%) [6]. The small variation between the two methods used for prevalence evaluation was mainly the result of the immigration of a seven-person family originating from Venezuela, in which four indigenous people were diagnosed with active tungiasis. Most likely, tungiasis endemicity is maintained in the Sanumás community by internal conditions and is subjected to very little influence from external immigration. In remote areas, although One Health actions should mainly be directed to internal community factors, a reintroduction of *T. penetrans* from external endemic communities must always be monitored.

The presence of *T. penetrans* in house soils was impressive and was confirmed by a direct method [6] as well as an innovative molecular method. However, the methods showed very poor agreement, which is probably a result of the presence of different species of fleas in the soil that cannot be differentiated via on-site visual analysis. The molecular method, although it can be more expensive, presents additional advantages, such as specificity once specific molecular targets can be designed, and soil samples can be easily stored and sent to reference labs, reducing the necessity of a specialized entomological evaluation of in-house soils. This can be a cost-effective strategy for vast territories.

At the basal evaluation, most of the dogs were infected by *T. penetrans* (83.95%; 95% CI = 74.12–91.17%). After a single administration of oral afoxolaner, no cases of canine tungiasis were reported. However, the mobility of the animals was very high. It was not possible to monitor how many dogs left the community or died. As the study protocol was aimed at the treatment of tungiasis in humans, systematic follow-up of each dog was not possible. There is no evidence to support the direct benefit of interventions related to domestic animals on the effectiveness of human tungiasis treatment. However, animals can be a source for soil infestation and a consequent human infection.

The acceptance of environmental and animal measures by the treated community is very important for the success of an action. Previous attempts to treat dogs with creolin baths described by the community were not well received by the indigenous populations. Concerns must also be raised regarding insecticide fumigation. Fipronil is a potent inhibitor of the invertebrate-specific glutamate-activated chloride channel [17] and is banned in many regions, including Europe, because of its long-term effect on insects [18]. The assistant team justified its use because of the tungiasis-related health emergency situation. Secondary complications, such as infections and mutilation (Figure 1 and Figure 7), were already present in the Sanumás community, and death resulting from complications of tungiasis had been reported in indigenous communities. The main concerns related to human contact with fipronil are chronic exposure and acute exposure to high concentrations of the product [17]. Monitoring soil infestation by direct examination and by molecular examination ensures that only the minimum necessary amount of insecticide is applied, reducing deleterious effects to the local fauna and flora and short-term and long-term exposure risks to human health.

As reported by previous clinical trials, the effectiveness and safety of topical dimethicone for the treatment of tungiasis is impressive [12,13]. In our study, from the subset of 362 indigenous people evaluated at 7 days after the application of NYDA^®^, only one patient with previous moderate tungiasis retained a unique active lesion. No side effects were reported in the short-term evaluation. The use of occlusive dressings after NYDA^®^ application showed no additional benefits once the cure rate was also very high in indigenous people who did not accept this additional procedure.

The long-term effectiveness of the application of topical dimethicone together with the One Health approach was also impressive. Only three indigenous people with tungiasis presented active lesions in the 1.5-year follow-up period, with two cases of recurrence. The very low number of cases detected after the start of the described One Health approach shows that joint measures taken to treat animals and the environment are effective in protecting indigenous people from reinfection. As reported in a recent systematic review, environmental control is a difficult but necessary task for clinical trials that aim to treat tungiasis [15]. In 2004, Heukelbach et al. (2004) used oral ivermectin to treat tungiasis, and similar results in the intervention and placebo groups were achieved [19]. The authors isolated trial participants in an area free of the infection source [19]. Recalling the physiopathology of tungiasis, a self-limited disease, we can expect that isolated control of the environment will also be effective. However, as topical dimeticone showed high effectiveness and a very low profile of adverse events, it is an interesting option for the treatment of human lesions.

This study suffers from the inherent limitations of any observational design, including the impossibility of treatment allocation randomization and impossibility of blinding indigenous people to the treatment received. However, it adds important information related to the effectiveness of the joint application of topical dimethicone and environmental control procedures that could not be evaluated in previous controlled trials. This study also provides information on effectiveness for a remote indigenous community, which can easily be applied to other communities.

## 5. Conclusions

We conclude that the control of human tungiasis with NYDA^®^, together with a single dose of afoxolaner for domestic dogs and a limited application of fipronil (controlled by direct and molecular methods of soil analysis), is an effective and safe measure for controlling tungiasis. This is an interesting strategy in epidemic clusters of tungiasis where complications such as secondary infection and mutilation occur. Longer follow-up times are mandatory to verify the effectiveness and safety of this strategy for the control of tungiasis on a large scale.

## Figures and Tables

**Figure 1 tropicalmed-08-00489-f001:**
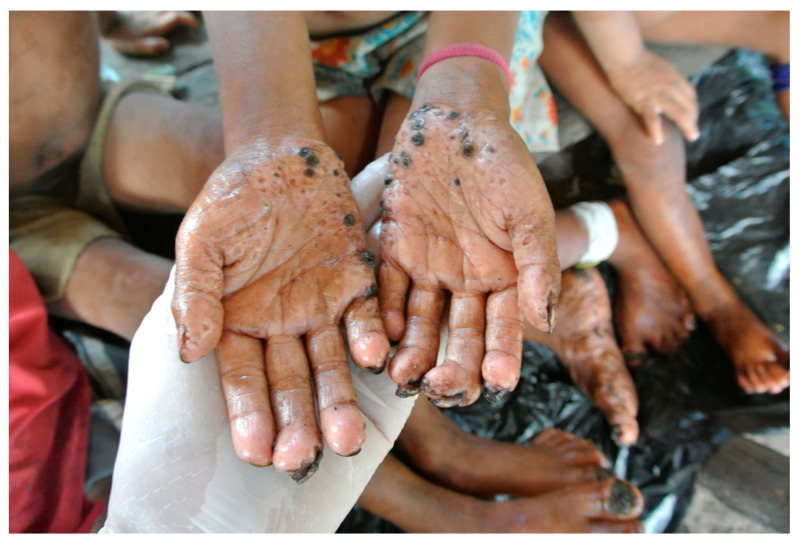
A case of tungiasis in a Sanumás community that resulted in permanent deformity of the distal interphalangeal joints of both hands.

**Figure 2 tropicalmed-08-00489-f002:**
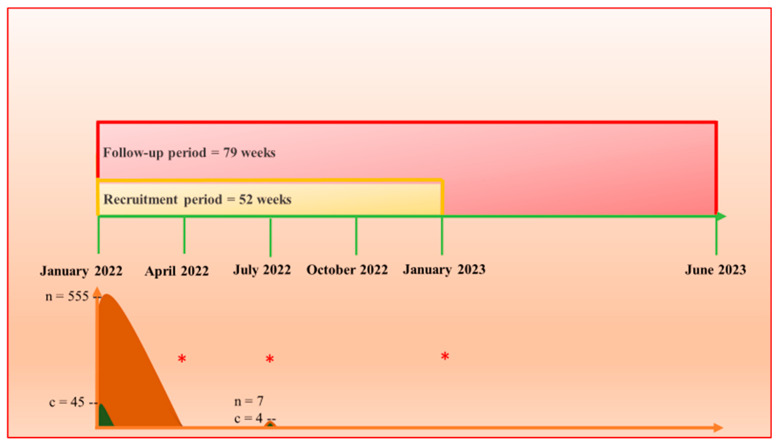
Graphical representation of the real-world cohort protocol divided into a 52-week recruitment period from January 2022 to January 2023 (yellow) and a 79-week follow-up period from January 2022 to June 2023 (red). n = total number of indigenous people (brown), c = number of cases of tungiasis (green), the red * represents each of the 3 tungiasis cases diagnosed during the follow-up period.

**Figure 3 tropicalmed-08-00489-f003:**
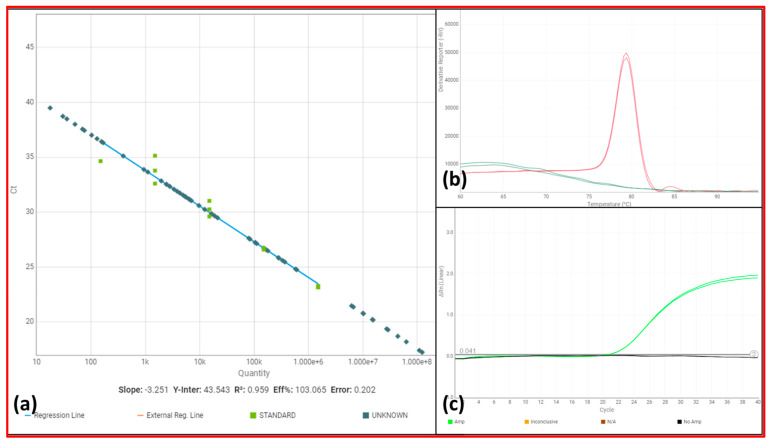
Images showing the standardization of the molecular analysis of soil infestation. The standard curve used cloned DNA at five dilution points starting at 1 million gene copies up to 100 gene copies. (**a**) The standard quantification curve aimed to test the quantitative capacity of the molecular target, showing that most samples were within the range of the positive controls (slope: −3.251; R^2^: 0.959; efficiency: 103.065); (**b**) melting curve analysis showing positive results (red) and negative controls (green); (**c**) linear amplification curves showing a positive result (green) and a negative result (black).

**Figure 4 tropicalmed-08-00489-f004:**
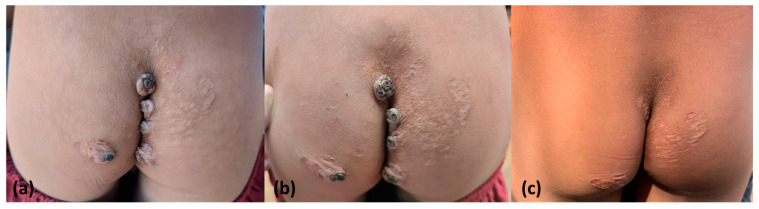
A case of coccygeal tungiasis before (**a**) and at 7 days (**b**) and 3 months after application of topical dimethicone NYDA^®^ (**c**). In the short-term evaluation (**b**), we observed that lesions presented dry crusts; in the long-term evaluation (**c**), all verrucous lesions had regressed in most cases.

**Figure 5 tropicalmed-08-00489-f005:**
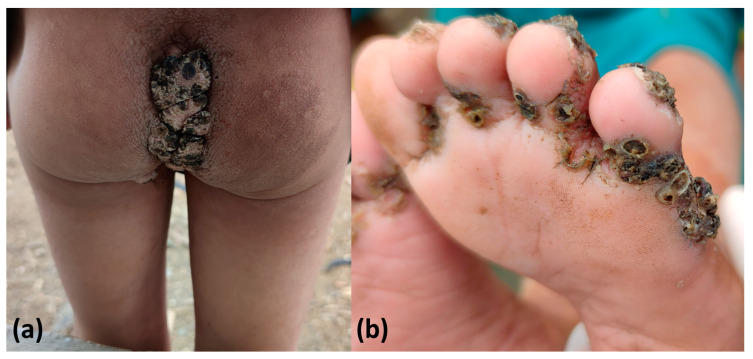
Details of coccygeal (**a**) and plantar (**b**) tungiasis lesions 7 days after application of NYDA^®^. The lesions became dry with abundant crusts.

**Figure 6 tropicalmed-08-00489-f006:**
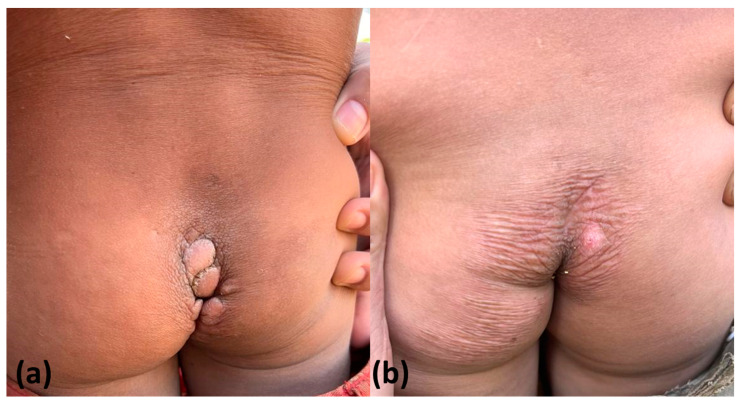
Details of coccygeal tungiasis (**a**) and (**b**) lesions at 3 months after application of NYDA^®^. Hypertrophic scars remain indefinitely in some cases (**a**).

**Figure 7 tropicalmed-08-00489-f007:**
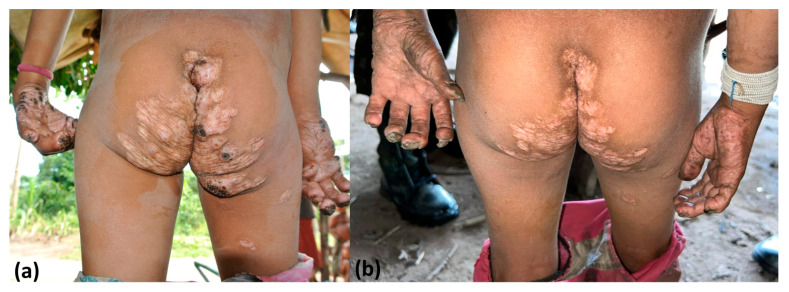
A case of severe tungiasis that resulted in permanent deformity of the distal interphalangeal joints of both hands before (**a**) and 3 months (**b**) after application of topical dimethicone NYDA^®^.

**Table 1 tropicalmed-08-00489-t001:** Univariate comparisons of the main demographic, clinical, animal, and environmental characteristics between indigenous people with and without tungiasis at the basal evaluation.

Variable	Tungiasis-Free Indigenous People (*n* = 513)	Tungiasis Indigenous People (*n* = 49)	*p* Value
Demographic factors			
Sex *n* (%)			
Female	247 (48.15%)	16 (32.65%)	0.054
Male	266 (51.85%)	33 (67.35%)	
Age *n* (%)			
Adults	274 (53.41%)	3 (6.12%)	<0.001
Children	239 (46.59%)	46 (93.88%)	
Living conditions			
Median number of houses in the community (IQR)	16 (2.00)	17 (2.00)	<0.001
Median number of inhabitants in the same house (IQR)	8 (4.00)	10 (4.00)	0.003
Median number of dogs living in the house (IQR)	1 (2.00)	1 (1.00)	0.299

*n* = total number of indigenous people; IQR = interquartile range.

## Data Availability

The data presented in this study are available upon request from the corresponding author.

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
