# Peer review of "The Effectiveness of Topical Dimethicone Together with a One Health Approach for the Control of Tungiasis in the Sanumás Communities, Yanomami Territory, Amazon Rainforest: A Real-World Study"

_tropicalmed, 2023, doi:10.3390/tropicalmed8110489_

Round 1

Reviewer 1 Report

Comments and Suggestions for Authors

In this interesting manuscript it is recognized that successful treatment of Tungiasis (sand flea, Tunga penetrans,  infestation) is highly dependent on adequate environmental control performed alongside the treatment of individual patients with topical medication. The need for a One Health integrated approach is discussed although the dog component of the work isn't fully examined. Were there any other animals e.g. pigs examined ? A semi-quantitative study is described outlining an observational cohort study designed to monitor the effectiveness of topical dimethicone together with environmental control of the flea in Sanumás communities of the, Amazon rainforest, Brazil. It is stated that 562 patients and a sub-sample of domestic dogs were followed for 1.5 years over 3-month intervals but the details of the dogs are not covered except as noted briefly in the discussion. Can this limitation also be mentioned in the materials and methods ? A new molecular method was used to evaluate the presence of fleas in soil samples, it is later stated that this improves detection of the target flea species. What other fleas are present ? It is stated that control of tungiasis was conducted by the Brazilian Ministry of Health and comprised topical dimethicone application (NYDA®) for humans and that a single-dose of oral afoxolaner was available to treat dogs where possible and in house soil fumigation was carried out using fipronil. There is reference to a previous study, could the findings of this previous study also be summarized briefly in this manuscript ?. Compliance with the supplementary use of dressings and other interventions was low, did this impact the efficacy of topical treatment ? It was concluded that the use of NYDA® together with animal and environmental interventions is effective in the control of tungiasis. This makes sense but there is no data presented to show the value of treating the animals ? Can the materials and methods and results sections be improved to address this ? The photographic figures should include photo credits, was permission granted to use these ? Figures 2 and 3 are a bit hard to read and should be improved. One key risk factor in Africa includes the use of unsealed flooring, has this been addressed in Brazil ? Hut design could then also be a risk factor to be addressed? Was there a risk assessment done for the regular use of fiprinol ? The sample size section 2.10 in the methods is a bit confusing, can this be improved ? Section 2.9, line 217 states that .'.............The main outcome .................the sentence is a bit confusing, is this a hypothesis or research question ? This needs to be re-worded. It might to be helpful to add some information about the flea life cycle and time frame to help support the study design and sampling framework.

Comments on the Quality of English Language

The materials and methods section can be improved, some sentences are misleading.

Author Response

Answer to reviewers

Dear Editor,

Thank you very much for the reconsideration of our points. Please find below detailed answers for all questions raised in the current review process.

Sincerely

Reviewer #1:

In this interesting manuscript it is recognized that successful treatment of Tungiasis (sand flea, Tunga penetrans,  infestation) is highly dependent on adequate environmental control performed alongside the treatment of individual patients with topical medication. The need for a One Health integrated approach is discussed although the dog component of the work is not fully examined. Were there any other animals e.g. pigs examined?

Despite previous reports that the Sanumás population raised animals such as pigs, cows and chickens, these animals were not present in the communities assessed. The explanation for this appears to be the intense social degradation of this population in recent years. No other animals were found during our evaluation. This is now explained in the text (lines 116 – 119).

A semiquantitative study outlining an observational cohort study designed to monitor the effectiveness of topical dimethicone together with environmental control of the flea in Sanumás communities of the Amazon rainforest, Brazil, is described. It is stated that 562 patients and a subsample of domestic dogs were followed for 1.5 years over 3-month intervals but the details of the dogs are not covered except as noted briefly in the discussion. Can this limitation also be mentioned in the materials and methods?

The limitations regarding the follow-up of all animals are now clarified in the text. As noted by reviewer #1, dogs are domestic animals that have a sort of nomad behavior meaning that they are not directly fed by their tutors, meaning that we could not find many of the dogs initially found in the first visit (lines 118 and 119).

A new molecular method was used to evaluate the presence of fleas in soil samples, and it was later stated that this method improves the detection of the target flea species. What other fleas are present?

In the soil evaluation, in addition to Tunga penetrams, we found a massive infestation by Ctenocephalides felis that also infested domestic dogs according to the veterinary evaluation. Although the direct examination on site is not capable of differentiating species, our molecular target is specific. This is now better explained in the text (Lines 387 - 390).

It is stated that control of tungiasis was conducted by the Brazilian Ministry of Health and comprised topical dimethicone application (NYDA®) for humans and that a single-dose of oral afoxolaner was available to treat dogs where possible and in house soil fumigation was carried out using fipronil. There is reference to a previous study, could the findings of this previous study also be summarized briefly in this manuscript?

As requested, an objective summary explaining the findings of the cited study was included in the introduction (lines 46 - 53).

Compliance with the supplementary use of dressings and other interventions was low, did this impact the efficacy of topical treatment?

We do not believe that this is a limitation for the treatment of tungiasis with NYDA, as our analytical approach did not show any difference regarding the use of occlusive dressing. Although the number of patients who accepted occlusive dressings was very low, the number of patients who did not use those dressings was large, and the result was also impressive. This is now better explained in the discussion section of the article (lines 462 - 464).

It was concluded that the use of NYDA® together with animal and environmental interventions is effective in the control of tungiasis. This makes sense but there is no data presented to show the value of treating the animals? Can the materials and methods and results sections be improved to address this?

We agree that there is no evidence to support the direct benefit of animal interventions on the effectiveness of tungiasis treatment. However, we reinforce the advantages of treating animals once they can be a source for soil infestation and a consequent human infection. Additionally, we have now highlighted that the medications given to animals are on label medications and that this was widely accepted by the community. This is now better explained in the text (lines 439 - 442).

The photographic figures should include photo credits, was permission granted to use these?

As informed in our submission form sent to the editorial team, all pictures were originally created by the authors. The consent form that patients signed included the acceptance of photographic documentation of lesions without identification of patients. This is in line with the Brazilian access to information law.

Figures 2 and 3 are a bit hard to read and should be improved.

We agree that those are complex images that may be difficult to interpret by the general audience, especially in figure 3, which shows advanced molecular techniques. These are original images showing the specificity of our designed molecular targets. We have improved the figure legends (Figures 2 and 3).

One key risk factor in Africa includes the use of unsealed flooring; has this been addressed in Brazil?

Although this is probably the case in Brazil, it has not been compared in indigenous populations. All house floors were made of compacted natural soil. This is now explained in the text (Lines 50 – 52).

Hut design could then also be a risk factor to be addressed?

We agree that this can be a risk factor. Hut design as a risk factor for tungiasis in this population was evaluated in a previous study that is now mentioned in the introduction section (lines 46 – 52).

Was there a risk assessment done for the regular use of fiprinol?

As this is an observational study, researchers had no influence on any procedure. However, we could monitor that the limited use of fipronil, a product with very small effects on mammals, was decided by a specialized team composed of entomologists. Health conditions of the population and soil and water monitoring are regularly performed by the Brazilian Ministry of Health. Our molecular analysis was useful for limiting the use of fipronil. This is now disclosed in the text (127 – 129).

The sample size section 2.10 in the methods is a bit confusing, can this be improved?

As requested, we have clarified our epidemiological procedures (Section 2.10).

Section 2.9, line 217 states that.’.............The main outcome.................the sentence is a bit confusing, is this a hypothesis or research question? This needs to be reworded. It might be helpful to add some information about the flea life cycle and time frame to help support the study design and sampling framework.

Although the definition of a main outcome and predictors comes from a hypothesis, this is a technical epidemiological procedure. For longitudinal studies, it is always important to define an outcome in the study protocol. This decision follows standard procedures needed by state-of-the-art research guidelines and is needed by the STROBE checklist. Chosing more than one outcome is frequently unfeasible because it is accompanied by the necessity of very large sample sizes. We decided that the presence or absence of tungiasis was our main outcome. This is now better explained in the text.

In addition, we agree that many technical considerations must come from the flea life cycle, but this may have very little influence on the registration of drugs and medical indications. The NYDA is still not registered in most endemic countries, and any study aiming to support registrations must choose as the main outcome the medical indication of the medication. This explains why we chose the presence of tungiasis as the main outcome. It also doesn’t mean that we can’t make additional analysis, however this analysis is less precise. This is now better explained in the text (Section 2.9).

Reviewer 2 Report

Comments and Suggestions for Authors

Dear Authors, thank you for submitting an interesting article entitled „Effectiveness of topical dimethicone together with a One Health approach for the control of tungiasis in the Sanumás communities, Yanomami territory, Amazon rainforest: a real-world study” because the transfer of information is always useful.

Below you can find some general and specific comments.

The paper investigates the effectiveness of topical dimethicone in conjunction with a One Health approach for controlling tungiasis in the Sanumás communities of the Yanomami territory in the Amazon rainforest, Brazil. The study conducted a real-world observational cohort analysis, including both human and animal subjects. The primary aim was to assess the occurrence of tungiasis after the utilization of topical dimethicone combined with the One Health approach.

The paper has a clear and relevant objective of controlling tungiasis in a challenging environment. The use of a One Health approach that considers both human and animal health is commendable.

Conducting a real-world observational cohort study is a strong point, as it simulates the practical application of treatments and interventions.

The study is highly relevant for the field of tropical medicine and public health, given the prevalence of tungiasis in the Amazon rainforest region.

The hypothesis that topical dimethicone, combined with the One Health approach, can effectively control tungiasis is testable and well-addressed in the study.

The conclusions drawn from the study appear consistent with the evidence presented. The combination of NYDA® with animal and environmental interventions seems to be effective in controlling tungiasis.

Author Response

Answer to reviewers

Dear Editor,

Thank you very much for the reconsideration of our points. Please find below detailed answers for all questions raised in the current review process.

Sincerely

Dear reviewer #2, we are very glad with the compliments. We have included some requests from reviewer #1 to clarify some specific procedures. We believe the article is now more didatic for the general public.
